# Experimental exploration of artificial intelligence and ADAMS simulation technology in the teaching of vertical hoop upward throw in rhythmic gymnastics

Qihong Ren⬤, Zongjue Ma⬤*

Physical Education Institute, Xinjiang Normal University, Urumqi, Xinjiang, China

* 1156414345@qq.com

## Abstract

Currently, rhythmic gymnastics teaching mainly focuses on traditional techniques and cannot break through issues such as blind spots in the coordination training between apparatus and body. In the academic field, research results are still dominated by single aspects such as new rules and event analysis. ADAMS technology can provide an innovative path to solve the above problems in rhythmic gymnastics teaching and scientific research. Based on the analysis of the advantages of ADAMS simulation technology and the theoretical basis for artificial intelligence technology to adapt to educational paradigms, this paper first uses ADAMS software to model the rhythmic gymnastics hoop through steps including model import, material definition, constraint relationship establishment, driving and force application, and simulation setting. It then conducts simulation of the vertical hoop thrown upward at two different angles (30 degrees and 45 degrees) and makes a comparative analysis of the counterclockwise and clockwise rotations of the vertical hoop thrown upward at the same angle. Furthermore, it carries out teaching experiments to explore the practical application of artificial intelligence technology in the teaching of upward throwing of the vertical hoop in rhythmic gymnastics, and draws the following conclusions: (1) The choice of projection angle has a differential impact on movement efficiency: a 30° projection angle is more suitable for basic standardized training due to its stable trajectory and moderate displacement, while a 45° projection angle is more conducive to enhancing artistic expression by virtue of its advantage in air retention. (2) Rotation direction has a significant impact on technical efficiency: clockwise rotation strengthens the spatiotemporal consistency of apparatus throwing and catching, while counterclockwise rotation optimizes the fluency of movement connection. Combined training can meet the requirements for innovative scoring. (3) ADAMS technology significantly improves teaching effectiveness through accurate trajectory prediction and mechanical analysis. Experimental data confirm that it is superior to traditional teaching methods in

**Data availability statement:** The relevant data have been uploaded to Figshare with the DOI: https://doi.org/10.6084/m9.figshare.29552849.

**Funding:** Teaching Research and Reform Program for Undergraduate Education of Xinjiang Autonomous Region Universities in 2023 (XJGXPTJG202361). Clarify that the paper was not funded by the faculty reform project, but was included as one of the outcomes of the project's completion.

**Competing interests:** The authors have declared that no competing interests exist.

terms of skill mastery, interest stimulation, and satisfaction. It is suggested that efforts should be made to construct an intelligent teaching closed-loop system based on ADAMS technology and promote a dual-track model of "scientific quantification-artistic expression", which is conducive to the scientific and intelligent development of physical education and art courses in colleges and universities.

## Introduction

The current teaching practice of rhythmic gymnastics faces the following key challenges: Technical skill instruction still relies primarily on experiential inheritance, lacking a quantitative analysis system based on biomechanical principles. There is an urgent need to shift away from subjective, experience-based teaching methods and strengthen the precise quantification of key technical parameters. Coordination training between apparatus and body movements focuses heavily on choreography, yet existing research data indicates that athletes' apparatus error rates are closely correlated with the lack of dynamic simulations of apparatus trajectories [1]. Currently, artificial intelligence technologies are being applied across various domains. For example, research on the integration of virtual simulation technology and sports has primarily focused on sports such as kayaking, skiing, and ball games. However, ADAMS technology can also provide an innovative approach to address the current issues in rhythmic gymnastics teaching. By establishing a rigid-flexible coupling system that includes both human skeletal-muscular models and rhythmic gymnastics apparatus, it is possible to numerically simulate key parameters such as the initial velocity of apparatus projection and rotational angular momentum. University curriculum reforms in this area can focus on three main aspects: (1) In body movement analysis, the software can create a dynamic model encompassing 12 major muscle groups, enabling the calculation of peak joint torques through inverse dynamics. (2) In apparatus dynamics, computational fluid dynamics algorithms can accurately simulate apparatus trajectories. (3) In injury prevention, by constructing a three-dimensional finite element model of the knee joint, it is possible to quantitatively analyze meniscal stress distribution during different landing actions.

From 2000 to 2020, research hotspots in rhythmic gymnastics mainly revolved around scoring rules, individual events, characteristics, competition and training, and aesthetic-difficulty sports [2]. The limited research on sports mechanics from this period was concentrated in the 1990s and primarily relied on video analysis and on-site observations, using theoretical frameworks such as the conversion of kinetic energy to thermal energy and Newton's First Law. Current research theories highlight three main points: (1) According to Web of Science data, studies on dynamic simulations of rhythmic gymnastics account for a small proportion in the field of sports engineering and are often limited to simple movement analyses. (2) There are significant disciplinary barriers in related research, with a low proportion of collaborative studies between sports institutions and engineering schools. (3) Existing virtual simulation research often employs simplified human models, such as simulating the

spine as a single rigid body, lacking in-depth studies that integrate apparatus and human movement. In university rhythmic gymnastics courses, the throwing and catching techniques of various apparatuses form the core of technically challenging routines.

The spatial orientation perception of learners is influenced by the mid-air directional changes of the apparatus. In particular, the coordination required between the hoop apparatus and body movements in hoop routines is particularly demanding [3]. Mastery of the vertical hoop upward throw technique, a hallmark of hoop routines, allows students to achieve harmony among power, amplitude, and execution speed, thereby demonstrating a certain level of professional skill. This paper uses ADAMS software, an artificial intelligence-based technology, to simulate and compare two parameters of the vertical hoop upward throw in rhythmic gymnastics: 30 degrees and 45 degrees. The results of this research not only establish a quantitative correlation model between the dynamic characteristics of sports apparatus and teaching parameters, providing a reusable interdisciplinary analysis framework for apparatus innovation research in sports engineering but also break through the traditional experience-driven decision-making model in rhythmic gymnastics teaching. By constructing a dynamic teaching feedback system using ADAMS simulation technology, a closed-loop optimization mechanism of "simulation prediction-movement correction-effect verification"is formed. The interactive training platform developed based on artificial intelligence technology can real-time analyze the mechanical characteristics of throwing movements and trajectory deviations of the apparatus, injecting digital elements into the reform of university rhythmic gymnastics courses.

## Materials and methods

### Virtual simulation method

The virtual simulation technology used in this study is based on the principles of multi-body system dynamics and implemented through ADAMS (Automatic Dynamic Analysis of Mechanical Systems) simulation software developed by Mechanical Dynamics.Inc. in the United States. As an internationally recognized standard tool for dynamic analysis of mechanical systems, this software can accurately simulate the dynamic behavior of complex motion chains by establishing parameterized multi-body system models. In this study, ADAMS software was used to model the multi-body dynamics analysis of the rhythmic gymnastics hoop, which mainly included the following steps: model import, material definition, constraint relationship setup, drive and force addition, and simulation configuration [4].

### Model import

In this study, a labeled rhythmic gymnastics hoop model was established in ADAMS software. According to the actual motion states and constraint relationships, as shown in Fig 1,constraint simulation was performed on the rhythmic gymnastics hoop [5], and marker points were automatically generated.

### Material definition of the rhythmic gymnastics hoop

As shown in Fig 2, the simulation involves a rhythmic gymnastics hoop, which is hollow with an inner diameter of 80 cm, a thickness of 18 mm, and a mass of 0.3 kg.

### Constraint relationships of the rhythmic gymnastics hoop

The rhythmic gymnastics hoop model imported into ADAMS software is analogous to a suspended object in a three-dimensional space, possessing six degrees of freedom: three translational degrees of freedom along the positive directions of the spatial coordinate axes and three rotational degrees of freedom around the coordinate axes [6]. In practical simulation, the rhythmic gymnastics hoop is assigned an initial velocity and rotation angle. However, relative to the global coordinate system, the overall rotational degrees of freedom of the hoop in the X, Y, and Z directions and its rotational

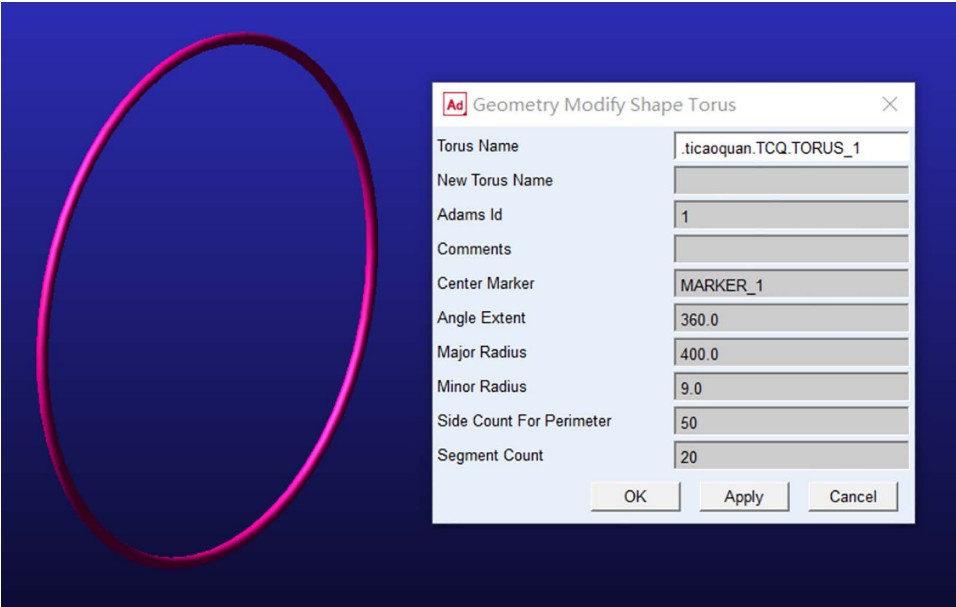

**Fig 1. The dynamic simulation model of instruments in rhythmic gymnastics hoop.**

freedom are not restricted, allowing it to fly freely in the air. The settings of each constraint pair for the rhythmic gymnastics hoop are shown in Table 1.

### Drive and force application

During simulation, it is assumed that the rhythmic gymnastics hoop is thrown with the same initial speed magnitude but different directions each time.

The loading parameters of the initial velocity after tossing at a 30-degree angle are shown in Fig 3: the total tossing velocity is 12 m/s, the velocity decomposed into the x-direction is 10,392 mm/s; and the velocity decomposed into the y-direction is 6,000 mm/s.

The loading parameters of the initial velocity after tossing at a 45-degree angle are shown in the Fig 4: the total tossing velocity is 12 m/s, the velocity decomposed into the x-direction is 8,485 mm/s; and the velocity decomposed into the y-direction is 8,485 mm/s.

### Simulation setup

The interactive simulation is primarily governed by two control parameters: simulation time and number of simulation steps. As shown in Fig 5, the simulation time is set to 1.8 seconds. To ensure the accuracy of simulation results while balancing simulation efficiency, the number of simulation steps is configured as 1000.

### Teaching experiment method

In this teaching experiment, the independent variable consisted of two teaching methods: the control group received traditional classroom instruction, while the experimental group experienced an integrated approach combining artificial intelligence technology with classroom teaching of the vertical hoop upward throw. The dependent variables were students' skill levels, situational interest in physical education, and training satisfaction. Prior to the experimental teaching, two

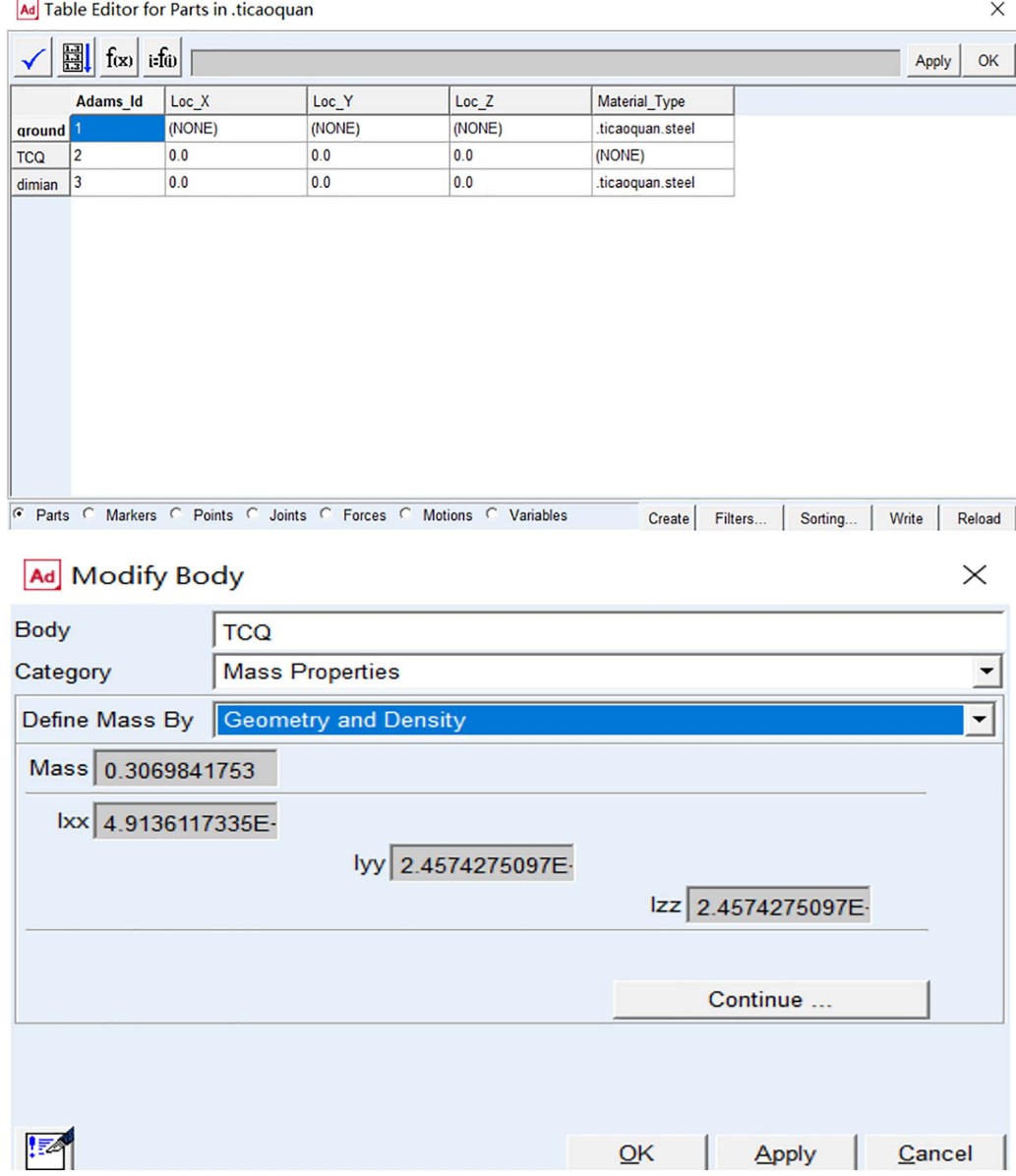

**Fig 2. Definition of rhythmic gymnastics hoop materials.**

**Table 1. Quantity and types of constraint pairs for rhythmic gymnastics hoop.**

| Constraint Type | Number of Constraint Pairs | Degrees of Freedom Restricted by Constraint Pairs |
|---|---|---|
| Revolute Joint | 0 | 5 |
| Fixed Joint | 0 | 6 |

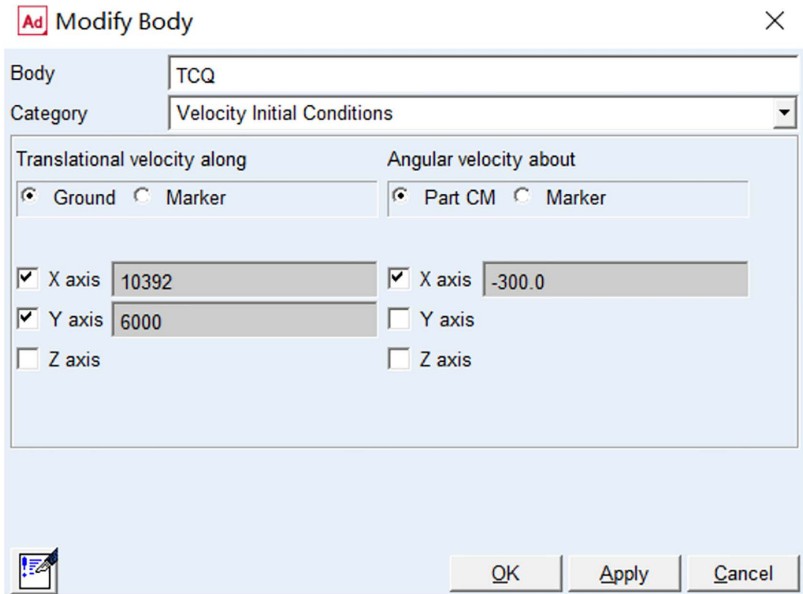

**Fig 3. Initial velocity loading parameters for a rhythmic gymnastics hoop thrown at 30 degrees.**

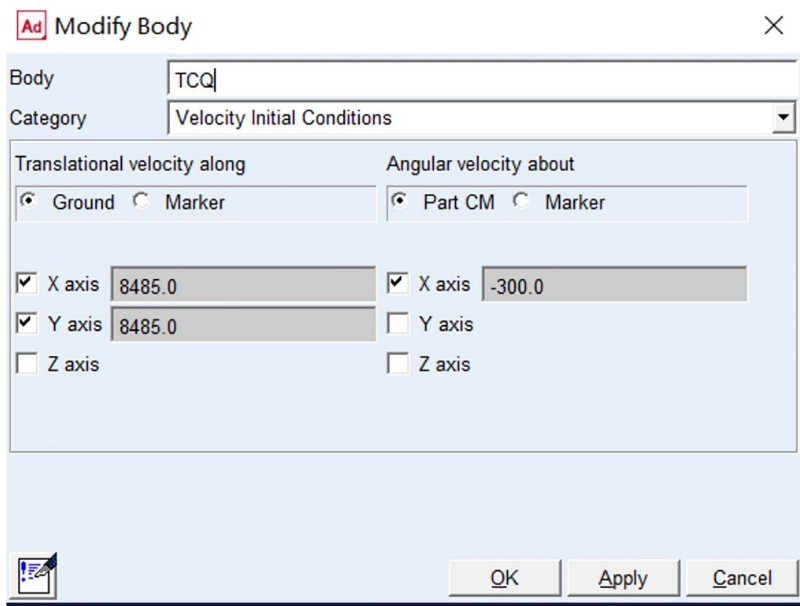

**Fig 4. Initial velocity loading parameters for a rhythmic gymnastics hoop thrown at 45 degrees.**

parallel classes from the same grade were randomly selected from the School of Physical Education at Xinjiang Normal University to serve as the experimental and control groups for the "vertical hoop upward throw" classroom instruction, with 27 students in each group. There were no significant baseline differences between the two groups of female students. At the end of the experimental teaching period, a single-blind evaluation method was employed for the assessment of

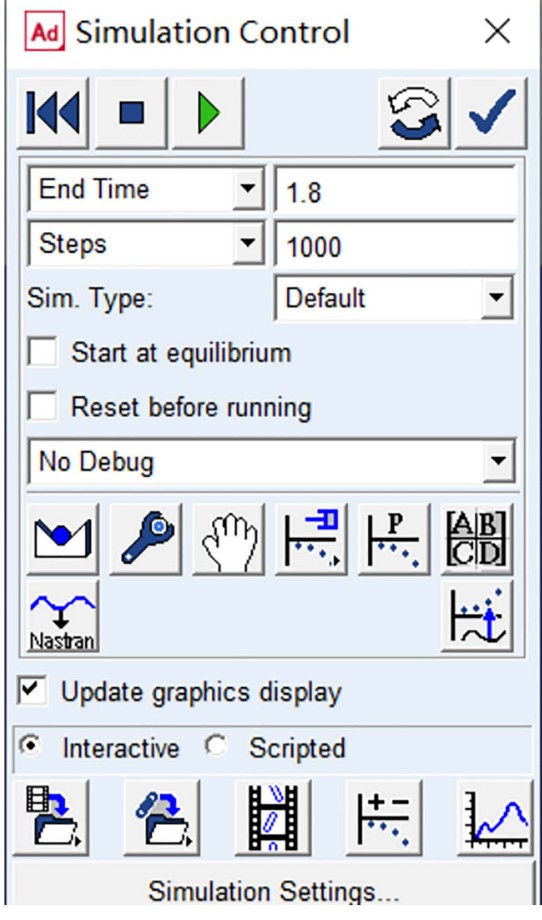

**Fig 5. Simulation time and step settings.**

the "vertical hoop upward throw" skill. First, students from both the experimental and control groups were mixed, and their assessment order was determined by lottery. Experts were then invited to conduct on-site evaluations. Additionally, Scale 25 and Scale 27 from the *Handbook of Psychological Scales in Sports Science* (2nd Edition, Zhang Liwei & Mao Zhixiong, 2010) [7] were used to assess the students' "situational interest in physical education" and "training satisfaction" through an online questionnaire administered via SoJump statistical analysis was performed using SPSS 25.0. The experiment was conducted from March 1, 2024, to June 30, 2024.

## Theoretical foundation

### The advantages of artificial intelligence ADAMS simulation technology

**The quickness of simulation operations.** ADAMS software is currently the most commonly used multi – body dynamics simulation software in the field of computer research. It can quickly import 3D models, rapidly identify and automatically create marker points. It can also quickly create constraint pairs, build a virtual prototype simulation platform, and input parameters to conduct simulation verification on 3D models [8]. The operational speed and convenience of ADAMS have led to its wide application in the research community.

**The simplification of working condition simulation.** The simplification of the ADAMS model is based on the software's algorithm mechanism, allowing the deletion or neglect of parts that have no or minimal impact on the kinematic and dynamic properties of the research object and research objectives. In this study, factors such as hoop belts and material wear can be ignored, demonstrating the simplification effect and facilitating simulation research.

**The applicability of simulation equivalence.** In ADAMS simulation, Boolean operations are performed on parts without relative motion and of the same material type to make them an integral whole, which can reduce the workload of material definition and constraint addition. The rhythmic gymnastics hoop is regarded as being composed of several parts. When multiple parts have no relative displacement in spatial geometry and are made of the same material with the same density, combining them into an integral whole is dynamically equivalent to the multi – component system in terms of kinetic properties.

### Theoretical basis for artificial intelligence technology adapting to educational paradigms

**Cognitive load theory.** Cognitive load theory posits that human cognitive structure consists of working memory (also known as short-term memory) and long-term memory. Working memory has limited capacity, while long-term memory has nearly unlimited capacity and serves as the core of learning. The theory classifies cognitive load into three types: intrinsic cognitive load, extraneous cognitive load, and germane cognitive load. It emphasizes the need to effectively utilize cognitive resources by reducing intrinsic and extraneous cognitive loads while increasing germane cognitive load to enhance learning outcomes [9]. Artificial intelligence technology helps students intuitively understand and master knowledge by simulating real environments, thereby reducing their extraneous cognitive load. Additionally, it provides customized teaching content based on students' individual needs, lowering their intrinsic cognitive load.

**Constructivism learning theory.** Constructivist learning theory emphasizes learners' initiative and the construction of personal experience. Its core idea is that students have the ability to actively learn and process information, and the task of educators is to provide appropriate educational environments and opportunities to stimulate students' learning interests and guide them to actively construct their own knowledge systems [10]. The artificial intelligence technology proposed in this paper can provide students with an interactive and exploratory skill-learning environment. According to students' learning progress and feedback, it offers timely teaching support and guidance to help them better understand and master technical knowledge, which involves constructing personal knowledge systems through simulated operations.

**Blended learning theory.** Blended learning theory originated from the reflection on the development dilemmas of e-Learning in foreign educational technology circles in 2001. It attempts to transform the traditional e-Learning model through blended learning, emphasizing the teaching philosophy of "teacher-led, student-centered." This theory integrates face-to-face teaching and online learning, aiming to achieve better learning outcomes by leveraging the leading role of teachers while reflecting students' initiative, enthusiasm, and creativity as the main body in the learning process [11]. Taking blended learning theory as one of the concepts for curriculum reform, this study combines artificial intelligence technology with traditional classroom teaching to form a blended learning model, giving play to the advantages of traditional teaching while using the convenience and interactivity of technology to improve teaching effectiveness.

## Results and discussion

### Simulation results

**Simulation of a vertical hoop thrown upward at 30°.**

(1) Clockwise rotation during 30° upward throw of the vertical hoop

As shown in Fig 6, the partial velocity curves in the X, Y, and Z directions indicate that: the velocity in the Z direction remains 0 throughout; the velocity in the X direction increases; the velocity in the Y direction decreases, drops to 0 after rising to the highest point, then accelerates in the reverse

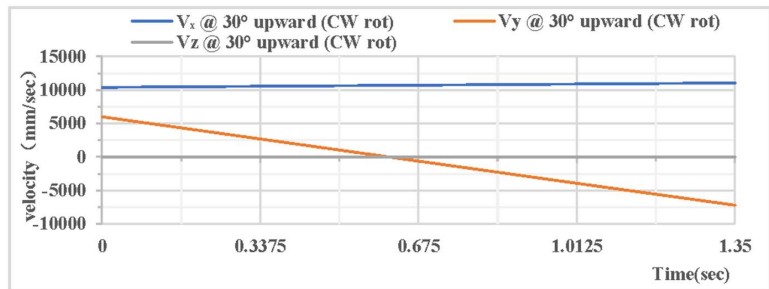

**Fig 6. Velocity components during 30° upward throw of a vertical hoop (CW rot).**

direction, and the velocity in the Y direction is −7.2 m/s upon landing. The combined velocity curves in the X, Y, and Z directions are shown in Fig 7. At the initial moment, the total velocity of the rhythmic gymnastics hoop is 12 m/s. At 0.56 s, the hoop rises to the highest point, with a velocity of 10.7 m/s at this moment. At the end of 1.35 s, the hoop lands, and the landing velocity is 13.2 m/s.

As shown in Fig 8, when the vertical hoop is thrown upward at 30 degrees with clockwise rotation, the curves of displacement components in the X, Y, and Z directions indicate that at the initial moment, the movement in the X direction is linear, the movement in the Y direction follows a parabolic trajectory, and the displacement in the Z direction is 0. As shown in Fig 9, when the vertical hoop is thrown upward at 30 degrees with clockwise rotation, the resultant displacement curve in the X, Y, and Z directions shows that at the initial moment, the total displacement of the rhythmic gymnastics hoop is 0 mm. When the instantaneous throwing velocity is 12 m/s, the hoop slides in the air for 1.35 seconds before landing, and the total displacement of the rhythmic gymnastics hoop after landing is 14498 mm.

(2) Counterclockwise rotation during 30° upward throw of the vertical hoop

As shown in Fig 10, the partial velocity curves in the X, Y, and Z directions indicate that: at the initial moment, the velocity in the Z direction remains 0 throughout; the velocity in the X direction increases; the velocity in the Y direction decreases, drops to 0 after rising to the highest point, then accelerates in the reverse direction, and the velocity in the Y direction is −7.1 m/s upon landing. As shown in Fig 11, the combined velocity curves in the X, Y, and Z directions show that: at the initial moment, the total velocity of the rhythmic gymnastics hoop is 12 m/s; the hoop apparatus rises to the highest point at 0.67 s, with a velocity of 10.8 m/s at this moment; the hoop lands at the end of 1.35 s, and the landing velocity is 12.1 m/s.

As shown in Fig 12, when the vertical hoop is thrown upward at 30 degrees with counterclockwise rotation, the curves of displacement components in the X, Y, and Z directions indicate that at the initial moment, the movement in the X direction is in a linear state, the movement in the Y direction follows a parabolic trajectory, and the displacement in the Z direction is 0. As shown in Fig 13, when the vertical hoop is thrown upward at 30 degrees with counterclockwise rotation, the resultant displacement curve in the X, Y, and Z directions shows that at the initial moment, the total displacement of the rhythmic gymnastics hoop is 0 mm. When the instantaneous throwing velocity of the hoop apparatus is 12 m/s, the rhythmic gymnastics hoop slides in the air for 1.35 seconds before landing, and the total displacement of the rhythmic gymnastics hoop after landing is 13609 mm.

(3) Comparative analysis of the 30° upward throw of the vertical hoop

When the vertical rhythmic gymnastics hoop is thrown at a 30° angle, the comparison curves of clockwise and counterclockwise rotations are shown in Fig 14: Among the velocity components in the X, Y, and Z directions, the

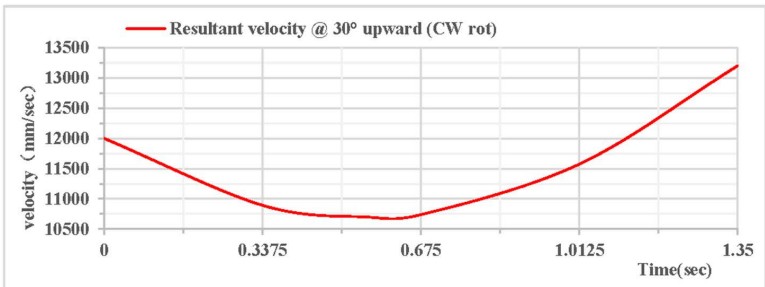

**Fig 7. Resultant velocity during 30° upward throw of a vertical hoop (CW rot).** $V_x$=X-direction velocity Component. Vy=Y-direction velocity component. Vz=Z-direction velocity component. @ = at. CW rot=Clockwise rotation.

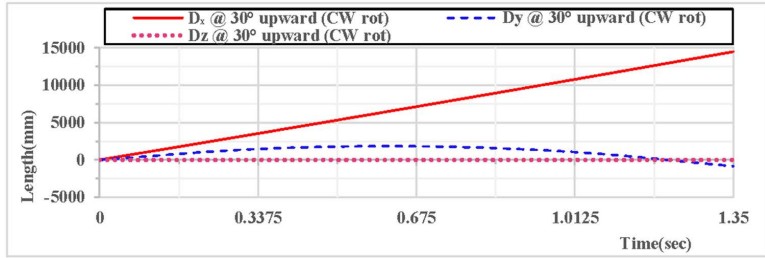

**Fig 8. Displacement components during 30° upward throw of a vertical hoop (CW rot).**

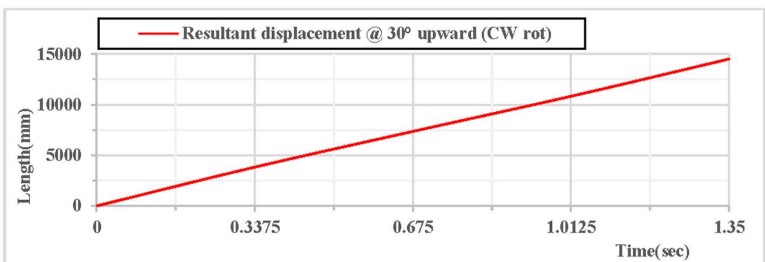

**Fig 9. Resultant displacement during 30° upward throw of a vertical hoop (CW rot).** $D_x$=X-direction displacement component. Dy=Y-direction displacement component. Dz=Z-direction displacement component. @ = at. CW rot=Clockwise rotation.

velocity in the Z direction is always 0, the velocity in the X direction increases, and the velocity in the Y direction decreases. As shown in Fig 15: When the hoop rotates counterclockwise, its overall velocity is relatively smaller, with a minimum velocity of 10080 mm/s; when the gymnastics hoop rotates clockwise, its minimum velocity is 10677 mm/s.

When the rhythmic gymnastics hoop is thrown at a 30° angle, the comparison curves of displacement between clockwise and counterclockwise rotations are shown in Fig 16: When the rhythmic gymnastics hoop rotates clockwise, its movement displacement is relatively larger, with a maximum displacement of 14,498 mm; when it rotates counterclockwise, the maximum displacement is 13,609 mm. The comparison of trajectory curves between clockwise and counterclockwise rotations is shown in Fig 17: After the rhythmic gymnastics hoop is thrown at a 30° angle, in the horizontal direction, the displacement of the hoop in clockwise rotation is larger than that in counterclockwise rotation, with a difference of 889 mm between the two.

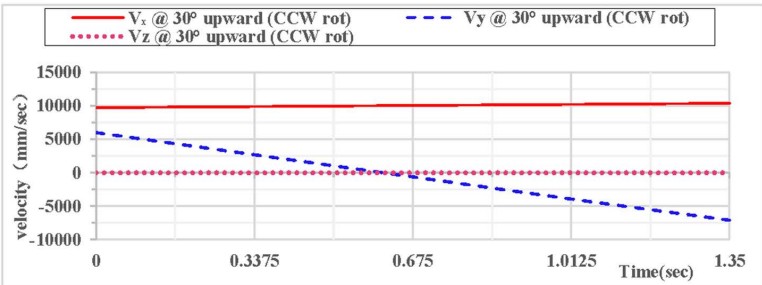

**Fig 10. Velocity components during 30° upward throw of a vertical hoop (CCW rot).**

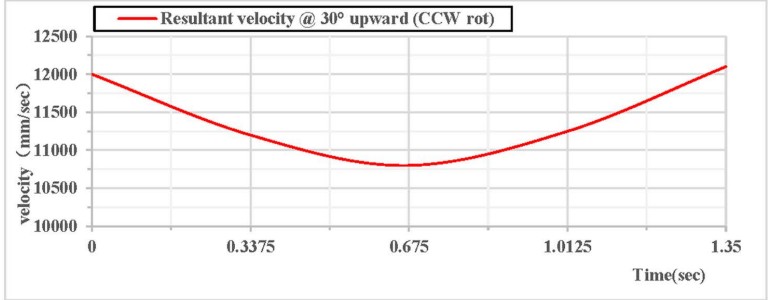

**Fig 11. Resultant velocity during 30° upward throw of a vertical hoop (CCW rot).** $V_x$ = X-direction velocity Component. Vy = Y-direction velocity component. Vz = Z-direction velocity component. @ = at. CCW rot = Counterclockwise rotation.

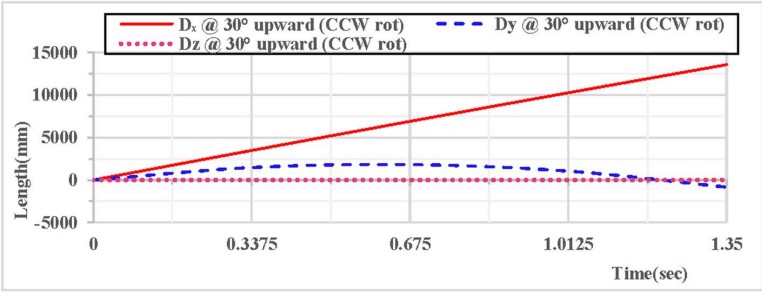

**Fig 12. Displacement components during 30° upward throw of a vertical hoop (CCW rot).**

### Simulation of a vertical hoop thrown upward at 45°.

(1) Clockwise rotation during 45° upward throw of the vertical hoop

As shown in Fig 18, the partial velocity curves in the X, Y, and Z directions indicate that: the velocity in the Z direction remains 0 throughout; the velocity in the X direction increases; the velocity in the Y direction decreases, drops to 0 after rising to the highest point, then accelerates in the reverse direction, and the velocity of the hoop apparatus in the Y direction is −9.1 m/s upon landing. As shown in Fig 19, the combined velocity curves in the X, Y, and Z directions show that: at the

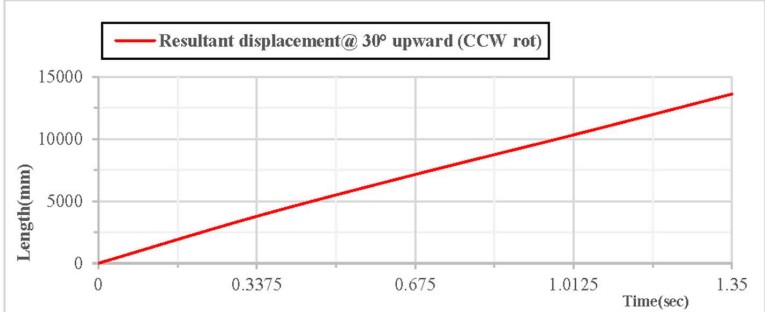

**Fig 13. Resultant displacement during 30° upward throw of a vertical hoop (CCW rot).** $D_x$ = X-direction displacement component. Dy = Y-direction displacement component. Dz = Z-direction displacement component. @ = at. CCW rot = Counterclockwise rotation.

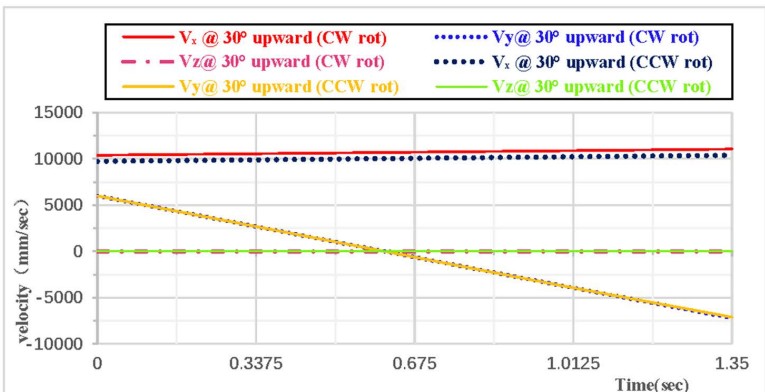

**Fig 14. Comparison of velocity components during 30° upward throw of a vertical Hoop(CW vs. CCW rot).**

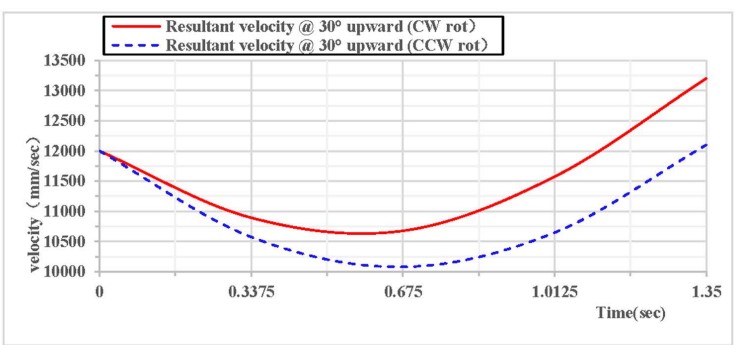

**Fig 15. Comparison of resultant velocity during 30° upward throw of a vertical Hoop(CW vs. CCW rot).** $V_x$ = X-direction velocity Component. Vy = Y-direction velocity component. Vz = Z-direction velocity component. @ = at. CW rot = Clockwise rotation. CCW rot = Counterclockwise rotation.

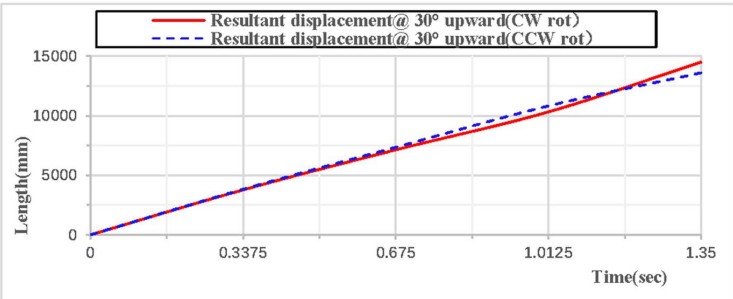

**Fig 16. Comparison of resultant displacement during 30° upward throw of a vertical Hoop(CW vs. CCW rot).**

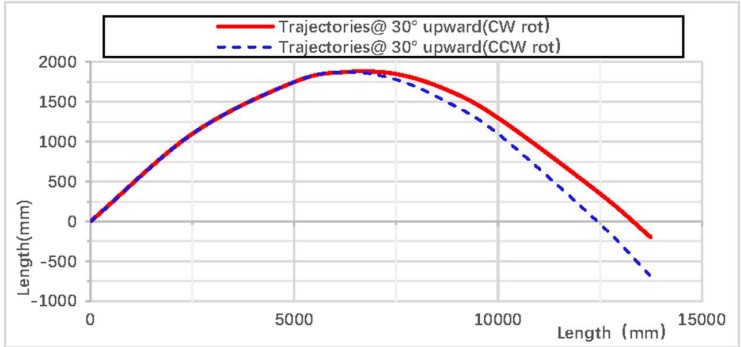

**Fig 17. Comparison of trajectories during 30° upward throw of a vertical Hoop(CW vs. CCW rot).** $D_x$ = X-direction displacement component. Dy = Y-direction displacement component. Dz = Z-direction displacement component. @ = at. CW rot = Clockwise rotation. CCW rot = Counterclockwise rotation.

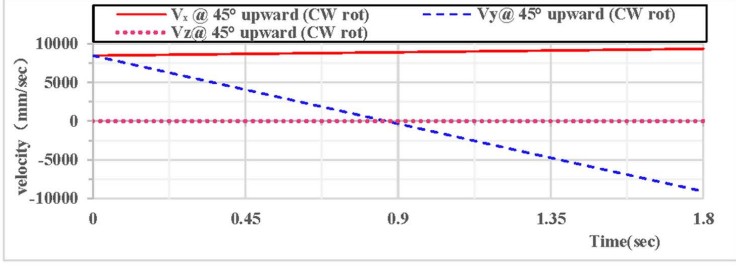

**Fig 18. Velocity components during 45° upward throw of a vertical hoop (CW rot).**

initial moment, the total velocity of the rhythmic gymnastics hoop is 12 m/s; the hoop apparatus rises to the highest point at 0.82 s, with a velocity of 8.9 m/s at this moment; the hoop lands at the end of 1.8 s, with a velocity of 13.1 m/s.

As shown in Fig 20, when the vertical hoop is thrown upward at 45 degrees with clockwise rotation, the displacement component curves in the X, Y, and Z directions indicate that the movement in the X direction is linear, the movement in the Y direction follows a parabolic trajectory, and the displacement in the Z direction is 0. As shown in Fig 21, when the vertical hoop is thrown upward at 45 degrees with clockwise rotation, the resultant displacement curve in the X, Y, and Z directions shows that at the initial moment, the total displacement of the rhythmic gymnastics hoop is 0 mm. When the

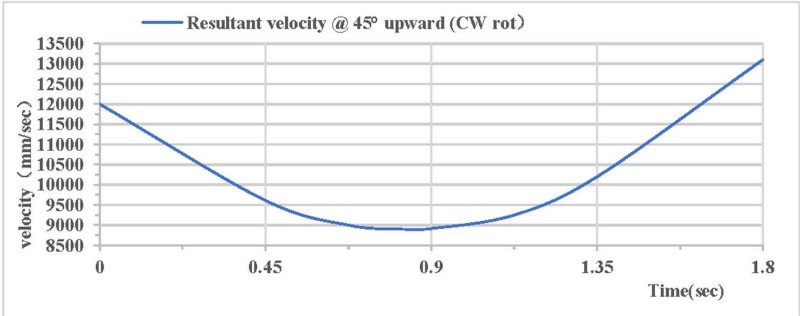

**Fig 19. Resultant velocity during 45° upward throw of a vertical hoop (CW rot).** $V_x$=X-direction velocity Component. Vy=Y-direction velocity component. Vz=Z-direction velocity component. @ = at. CW rot=Clockwise rotation.

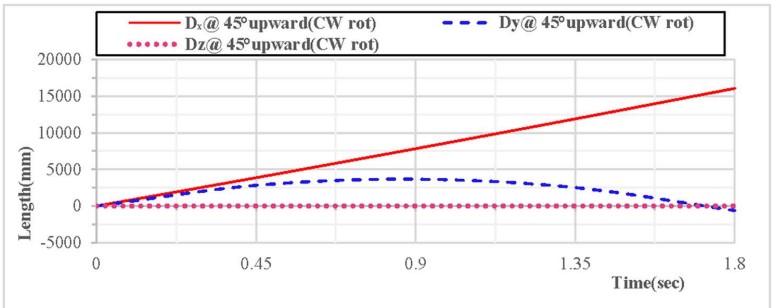

**Fig 20. Displacement components during 45° upward throw of a vertical hoop (CW rot).**

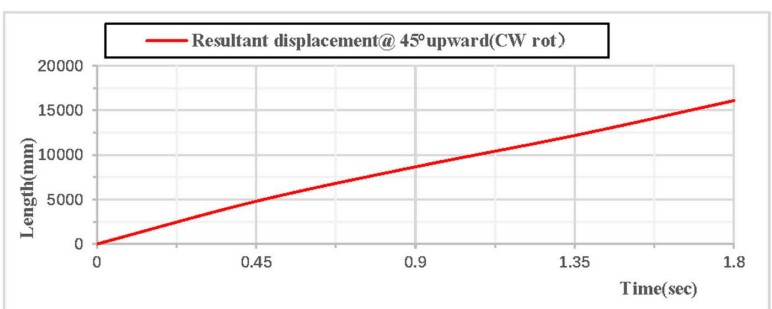

**Fig 21. Resultant displacement during 45° upward throw of a vertical hoop (CW rot).** $D_x$=X-direction displacement component. Dy=Y-direction displacement component. Dz=Z-direction displacement component. @ = at. CW rot=Clockwise rotation.

instantaneous throwing velocity is 12 m/s, the hoop apparatus glides in the air for 1.8 seconds before landing, and the total displacement of the rhythmic gymnastics hoop after landing is 16076 mm.

(2) Counterclockwise rotation during 45° upward throw of the vertical hoop

As shown in Fig 22, the partial velocity curves in the X, Y, and Z directions indicate that: the velocity in the Z direction remains 0 throughout; the velocity in the X direction increases; the velocity in the Y direction decreases, drops to 0 after

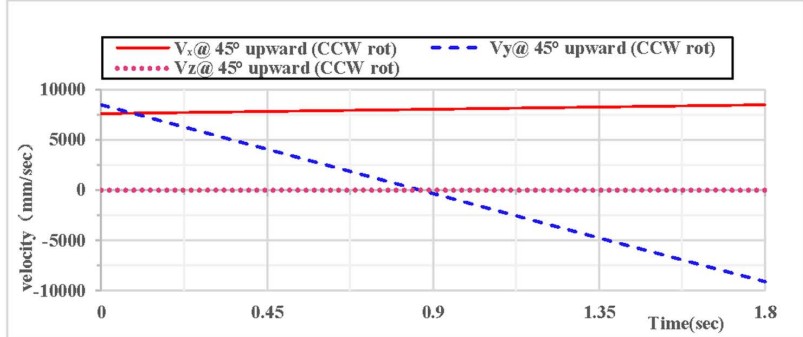

**Fig 22. Velocity components during 45° upward throw of a vertical hoop (CCW rot).**

rising to the highest point, then accelerates in the reverse direction, and the velocity of the hoop apparatus in the Y direction is −9.1 m/s upon landing. As shown in Fig 23, the combined velocity curves in the X, Y, and Z directions show that: at the initial moment, the total velocity of the rhythmic gymnastics hoop is 12 m/s; the hoop apparatus rises to the highest point at 0.91 s, with a velocity of 8.1 m/s at this moment; the hoop lands at the end of 1.8 s, and the landing velocity is 11.9 m/s.

As shown in Fig 24, when the vertical hoop is thrown upward at 45 degrees with counterclockwise rotation, the displacement component curves in the X, Y, and Z directions indicate that the X direction is in a state of linear motion, the Y direction follows a parabolic trajectory, and the Z direction is 0. As shown in Fig 25, when the vertical hoop is thrown upward at 45 degrees with counterclockwise rotation, the resultant displacement curve in the X, Y, and Z directions shows that the total displacement of the rhythmic gymnastics hoop is 0 mm at the initial moment. When the instantaneous throwing velocity is 12 m/s, the rhythmic gymnastics hoop glides in the air for 1.8 seconds before landing, and the total displacement of the rhythmic gymnastics hoop after landing is 14494 mm.

(3) Comparative analysis of the 45° upward throw of the vertical hoop

When the rhythmic gymnastics hoop is thrown at a 45-degree angle, the comparison curves of clockwise and counterclockwise rotations are shown in Fig 26: Among the velocity components in the X, Y, and Z directions, the velocity in the Z direction is always 0, the velocity in the X direction increases, and the velocity in the Y direction decreases. As shown in Fig 27: When the hoop rotates counterclockwise, its overall velocity is relatively smaller, with a minimum velocity of 8052 mm/s; when the gymnastics hoop rotates clockwise, its minimum velocity is 8896 mm/s.

When the rhythmic gymnastics hoop is thrown at a 45° angle, the comparison curves of displacement between clockwise and counterclockwise rotations are shown in Fig 28: When the rhythmic gymnastics hoop rotates clockwise, its movement displacement is relatively larger, with a maximum displacement of 16076 mm, while when the rhythmic gymnastics hoop rotates counterclockwise, its maximum displacement is 14494 mm. The comparison of trajectory curves between clockwise and counterclockwise rotations is shown in Fig 29: After the rhythmic gymnastics hoop is thrown at a 45° angle, in the horizontal direction, the displacement of the rhythmic gymnastics hoop in clockwise rotation is larger than that in counterclockwise rotation, with a difference of 1582 mm between the two.

## Discussion

### Classroom teaching techniques for 30° and 45° upward throws of the vertical hoop

There are four teaching techniques for the upward throw of the vertical hoop: (1) During the preparation process before throwing the hoop upward, the hoop must not touch the ground and should keep swinging at the side of the body; (2)

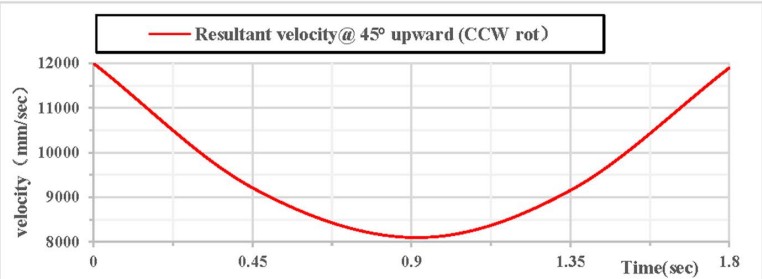

**Fig 23. Resultant velocity during 45° upward throw of a vertical hoop (CCW rot).** $V_x$=X-direction velocity Component. Vy=Y-direction velocity component. Vz=Z-direction velocity component. @ = at. CCW rot=Counterclockwise rotation.

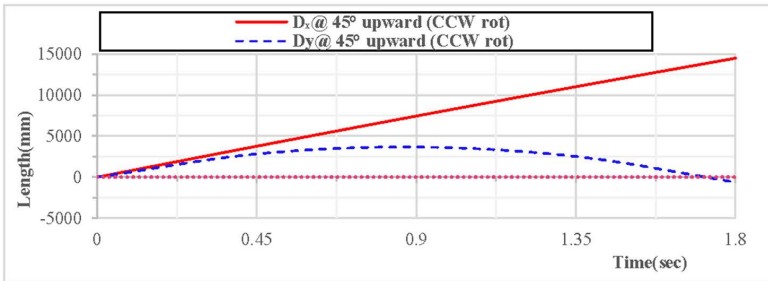

**Fig 24. Displacement components during 45°upward throw of a vertical hoop (CCW rot).**

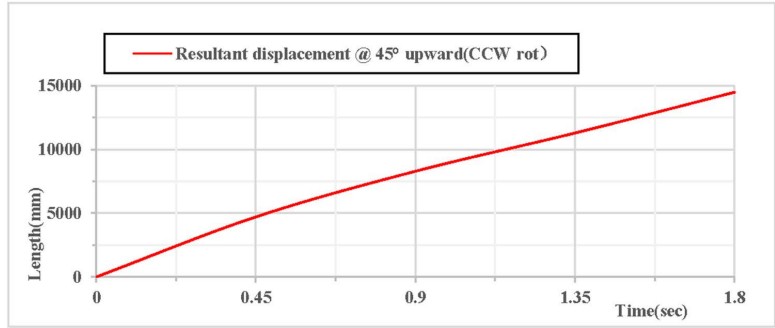

**Fig 25. Resultant displacement during 45°upward throw of a vertical hoop (CCW rot).** $D_x$=X-direction displacement component. Dy=Y-direction displacement component. Dz=Z-direction displacement component. @ = at. CCW rot=Counterclockwise rotation.

When the hoop is rotating, the palm and back of the hand should be closely attached to the inner edge of the hoop. At the moment the hoop spins upward, the right arm should extend upward from a bent elbow position to throw the hoop forward and upward; (3) The arm used to throw the hoop should stretch forward and upward, with the fingertips rotating upward, and the palm should be inserted forward into the rear lower edge of the hoop, continuing to rotate forward along with the inertial rotation of the hoop; (4) The movement of pushing off the ground and the swinging of the arm should be coordinated and consistent, exerting force together.

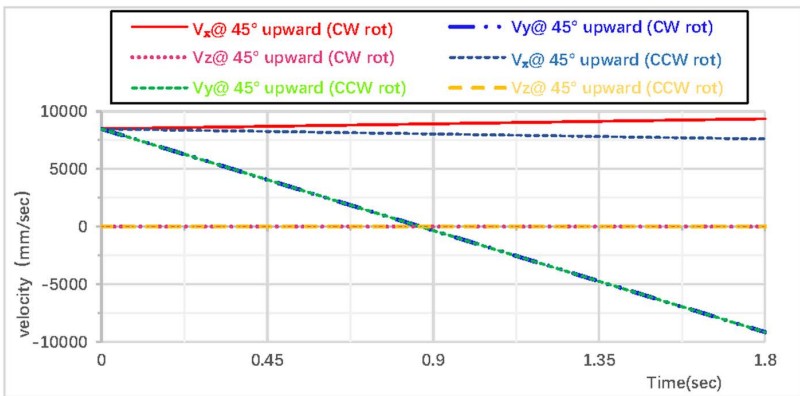

**Fig 26.  Comparison of velocity components during 45° upward throw of a vertical Hoop(CW vs. CCW rot).**

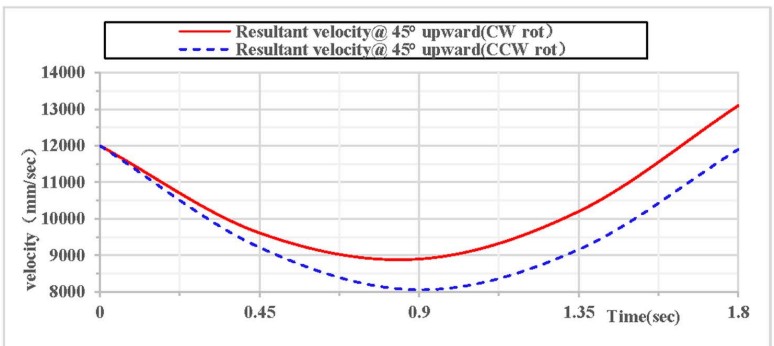

**Fig 27.  Comparison of resultant velocity during 45° upward throw of a vertical Hoop (CW vs. CCW rot).** $V_x$ = X-direction velocity Component. Vy = Y-direction velocity component. Vz = Z-direction velocity component. @ = at. CW rot = Clockwise rotation. CCW rot = Counterclockwise rotation.

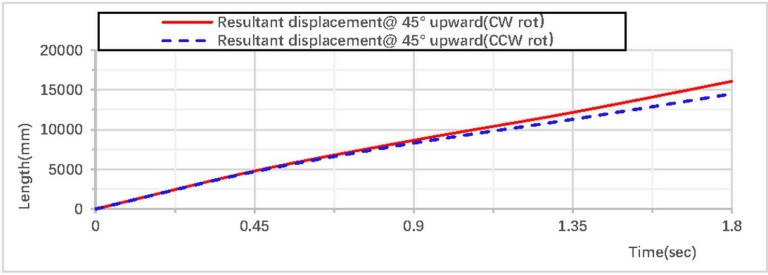

**Fig 28.  Comparison of resultant displacement during 45° upward throw of a vertical Hoop (CW vs. CCW rot).**

The teaching focus for the 30° and 45° upward throws of the vertical hoop lies in the following: when throwing the hoop, the arm must be straight, the release of the hoop should be slightly delayed, and the wrist should flick upward gently to make the hoop rotate in the air. The hoop must remain perpendicular to the ground throughout its flight, following a parabolic trajectory. During the teaching process, teachers need to calmly explain the key points of each difficult movement

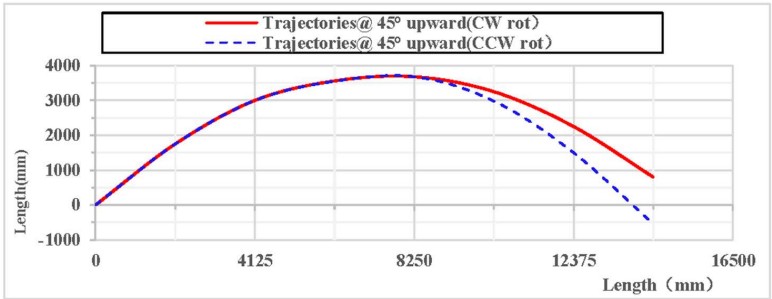

**Fig 29. Comparison of trajectories during 45° upward throw of a vertical Hoop (CW vs. CCW rot).** $D_x$ = X-direction displacement component. Dy = Y-direction displacement component. Dz = Z-direction displacement component. @ = at. CW rot = Clockwise rotation. CCW rot = Counterclockwise rotation.

clearly and accurately, highlighting the priorities. Then, students are asked to practice throwing and catching the hoop forward and upward, with the speed gradually increasing. The exercise load should be gradually intensified from stationary practice to walking and then running practice [12].

The integration teaching of rhythmic gymnastics apparatuses with various body movements is a difficult point in rhythmic gymnastics teaching. In hoop apparatus teaching, whether it is the double cooperation of the 30° upward throw of the vertical hoop or the difficult coordination between the 45° upward throw of the vertical hoop and body movements, it is necessary to start from basic techniques. When practitioners can perform the skillful movements freely, they can enter the links of students' innovative cooperation and display evaluation. At this time, the teaching focus should shift from teachers' "teaching" to students' "learning", to prevent students from becoming mentally slack and entering the fatigue period too early. This is not just a simple practice of body movements, but a complex teaching process where practitioners practice with light apparatuses accompanied by music. Teachers need to teach students in accordance with their aptitude based on the teaching difficulty of rhythmic gymnastics hoop apparatuses, superimpose online teaching technology with various teaching methods, to stimulate students' practical interest in the apparatus movements of this course, tap into students' initiative and enthusiasm for learning, effectively deepen practical experience, and meet the internal requirements of improving students' innovative awareness [13].

### Kinematical characteristics and teaching strategies of the 30° projection angle for the vertical hoop

When projected at 30 degrees, the total displacement of the vertically oriented hoop with clockwise rotation is 14,498 mm, while that with counterclockwise rotation is 13,609 mm, resulting in a horizontal displacement difference of 889 mm. The resultant velocity drops to 10.7 m/s at the highest point at 0.56 seconds and increases to 13.2 m/s upon landing, showing a dynamic characteristic of "slow ascent - rapid descent". Both the velocity and displacement of the apparatus with clockwise rotation are superior to those with counterclockwise rotation, which may be related to human habits, right-handed individuals being more adaptable to clockwise movements, and the rotational inertia of the apparatus. The moderate airtime of 1.35 seconds for the 30-degree upward projection is suitable for connecting short-term ground movements, such as catching the hoop after a single-leg spin, and its advantage in horizontal displacement can enhance the artistic expressiveness of the movements.

The key points in technical training for the 30 degrees upward throw of the vertical hoop are as follows: Firstly, the rhythm of throwing and catching. By visualizing the velocity curve, emphasis is placed on adjusting body posture near the highest point, such as arm extension for balance. Secondly, adaptation to rotation direction. Specialized training is designed according to students' dominant hands; for example, right-handed students are prioritized to practice clockwise projection to reduce the difficulty of movement learning.

## Kinematic characteristics and teaching strategies of the 45° projection angle for the vertical hoop

Total displacement significantly increases at 45 degrees projection (16,076 mm for clockwise vs. 14,494 mm for counterclockwise), but airtime extends to 1.8 s, with slightly reduced landing velocity (13.1 m/s for clockwise vs. 11.9 m/s for counterclockwise). The minimum velocity for clockwise projection (8,896 mm/s) is higher than counterclockwise (8,052 mm/s), indicating that the apparatus rotation direction affects energy loss. -The long airtime at 45 degrees is suitable for complex aerial skills (e.g., two mid-air twists to catch the hoop), and its large displacement can be used to design long-distance movement combinations with wide spans.

Key Training Focus for 45-Degree Vertical Hoop toss Technique: (1)Energy Efficiency Optimization: Use counterclockwise projection training to enhance students' perception of apparatus rotational resistance and strengthen explosive force control in tossing movements. (2)Hoop-Catching Stability: Design special cushioning technique exercises based on landing velocity differences.

## Balancing projection angles and artistic expressiveness

In the teaching practice of projectile sports events, the dynamic balance between technical parameters and artistic expressiveness represents the core challenge in teaching design. Preventing the excessive pursuit of maximum displacement, which may cause the range of motion to exceed reasonable limits, not only increases the difficulty of body control but also disrupts the smooth articulation of the movement chain. While solely relying on the speed advantage in the rotational direction can ensure technical stability, it undermines the creative space for movement choreography and leads to the formation of rigid performance patterns [14].This contradictory relationship between technical parameters and artistic expression essentially reflects the collaborative challenge between kinetic elements and aesthetic elements in sports training.

Teaching Strategies Should Adopt Stage-Based Objectives:

(1) Foundation Stage:Use 30-degree projection to establish movement standardization, leveraging its moderate displacement to reduce cognitive load. The trajectory at this angle balances appropriate displacement and controllable rotation speed, allowing learners to achieve an optimal balance between movement standardization and energy consumption.

(2) Advanced Stage: Introduce 45-degree projection to challenge complex choreography, combining clockwise and counterclockwise rotations to enhance artistic expression levels. Increasing parabolic curvature can unlock movement potential, and alternating clockwise-counterclockwise rotation training can develop multiple basic variant movements, significantly improving the spatio-temporal complexity of movement combinations.

(3) Virtual-Reality Integrated Training: Consider integrating simulation data with wearable sensors to provide real-time feedback on projection angle and rotation speed, assisting students in balancing technical precision and artistic expression. This quantified training model can offer a new paradigmatic path for the scientific training of sports aesthetics.

## Experimental results

**Analysis of skill assessment results of students in the control group and experimental group for the teaching of upward thrown vertical hoop.** From the data in Table 2, it can be seen that the average skill assessment score of students in the experimental group is 78.190, which is significantly higher than 72.480 in the control group, indicating that the teaching method of the experimental group is more effective. The standard deviation of the control group is 5.659, while that of the experimental group is 1.272, showing that the scores of the experimental group are more concentrated with higher consistency. The results of the independent samples t-test show that the t-value is −5.109 and the P-value is 0.000 (P ≤ 0.001), which further confirms that the difference between the two groups is statistically significant, indicating that the teaching method of the experimental group is significantly better than that of the control group in improving students' skill assessment.

**Table 2. Statistical results of skill assessment for students in the control and experimental group in the teaching of upward vertical hoop (N = 54).**

| G | SS | M | SD | t | P |
|---|---|---|---|---|---|
| control group | 27 | 72.480 | 5.659 | −5.109 | 0.000*** |
| experimental group | 27 | 78.190 | 1.272 | | |

G = Group; SS = Sample size; M = Mean; SD = Standard Deviation; t = statistic of t-test; P = Probability; P-value of ≤0.05 (*) indicates a significant difference; P-value of ≤0.01 (**) denotes a very significant difference; P-value of ≤0.001 (***) signifies an extremely significant difference.

**Analysis of the results regarding sports situational interest among students in both the control and experimental groups.** The data in Table 3 shows that the average score of students in the control group for situational interest in physical education is 96.480, while that of students in the experimental group is 101.070. Students in the experimental group demonstrate a high level of situational interest in physical education, indicating that the innovative teaching model in the experimental group can stimulate students' interest and enthusiasm. The standard deviation of the control group is 6.647, while that of the experimental group is 8.629, meaning the variability of scores on situational interest in physical education among students in the experimental group, i.e., the degree of dispersion in data distribution, is greater than that in the control group. The t-test value is −2.191, and the P value is 0.033 ($p \leq 0.05$), which indicates that there is a significant difference in situational interest in physical education between the control group and the experimental group in the teaching of vertical circle throwing upward.

**Analysis of satisfaction outcomes for students in the control and experimental groups.** According to the data in Table 4, the average score of students' satisfaction in the control group is 28.444, while that in the experimental group is 32.410. The experimental group scores higher than the control group, indicating that the experimental group students' needs and expectations for professional course teaching have been significantly improved. The standard deviations of the two groups are not much different, being 5.639 and 5.264 respectively, which shows that the distribution of students' satisfaction in the two groups is relatively consistent. The t-test value is −2.669, and the P value is 0.010 ($P \leq 0.01$), which indicates that there is a very significant difference in training satisfaction between the control group and the experimental group in the teaching of vertical circle throwing upward.

## Discussion

**Discussion on students' professional skills.** In rhythmic gymnastics teaching, the standardization, coordination, and precision of technical movements play a decisive role in learning outcomes. This study focuses on the technical movement of "upward-thrown vertical hoop" by comparing the differences between traditional teaching and digitally-empowered teaching, revealing the role of technical means in improving teaching effectiveness. The results of the independent samples T-test in Table 2 (t = −5.109, P = 0.000) further verify the extremely significant difference between the two groups, highlighting the effectiveness of technology-empowered teaching. Current rhythmic gymnastics teaching generally relies on teachers' demonstrations and students' imitation, but it is difficult to accurately convey detailed movement details such as hoop-throwing angles and physical coordination through traditional methods. The innovative practice of the experimental group shows that 3D dynamic decomposition technology can break down movements into visual modules in the temporal and spatial dimensions, helping students establish a clear cognitive representation of movements. The real-time data feedback system provides objective evaluation through quantitative indicators such as hoop-throwing height and trajectory stability, prompting students to adjust their practice strategies in a targeted manner. In addition, the interesting design of digital tools can stimulate students' motivation for independent practice, extend effective training time, and thus accelerate the internalization of skills.

**Table 3. Statistical results of sports situational interest among students in the control and experimental group in the teaching of upward vertical hoop(N = 54).**

| G | SS | M | SD | t | P |
|---|---|---|---|---|---|
| control group | 27 | 96.480 | 6.647 | −2.191 | 0.033** |
| experimental group | 27 | 101.070 | 8.629 | | |

G = Group; SS = Sample size; M = Mean; SD = Standard Deviation; t = statistic of t-test; P = Probability; P ≤ 0.05 (*) indicates significant difference; P ≤ 0.01 (**) indicates highly significant difference; P ≤ 0.001 (***) indicates highly significant difference.

**Table 4. Statistical results of training satisfaction among students in the control and experimental group in the teaching of upward vertical hoop(N = 54).**

| G | SS | M | SD | t | P |
|---|---|---|---|---|---|
| control group | 27 | 28.444 | 5.639 | −2.669 | 0.010*** |
| experimental group | 27 | 32.410 | 5.264 | | |

G = Group; SS = Sample size; M = Mean; SD = Standard Deviation; t = statistic of t-test; P = Probability; P ≤ 0.05 (*) indicates significant difference; P ≤ 0.01 (**) indicates highly significant difference; P ≤ 0.001 (***) indicates highly significant difference.

However, the universality of technological intervention still needs further discussion. For example, advanced complex movements such as apparatus exchanges and multi-axial rotations may have different requirements for technical tools; in teaching scenarios, teachers' technical application capabilities and teaching design levels may also affect the effectiveness. Future research could expand the sample size and explore hybrid teaching models, which can retain the advantages of teachers' personalized guidance in traditional teaching while using technical tools to strengthen movement analysis and process monitoring, thus building a more efficient rhythmic gymnastics teaching system. Additionally, attention should be paid to the potential impact of technical tools on students' creativity and artistic expressiveness to avoid excessive reliance on data-driven indicators leading to mechanized movements.

**Discussion on students' interest in physical education contexts.** In rhythmic gymnastics teaching, interest in physical education contexts, as a crucial component of intrinsic motivation, directly influences students' participation and skill acquisition. By comparing the impacts of traditional and innovative teaching models on students' interest in physical education contexts, this study reveals the significant advantages of technology-empowered teaching in stimulating learning motivation. As shown in Table 3, the mean score of the experimental group in physical education context interest(101.070) was significantly higher than that of the control group (96.480), with a statistically significant difference(t = −2.191, P = 0.033). This result indicates that the digital and interactive teaching methods adopted by the experimental group can effectively enhance students' interest levels, providing empirical evidence for the reform of rhythmic gymnastics teaching.

Current rhythmic gymnastics teaching commonly faces issues such as monotonous training and delayed feedback, where over-reliance on mechanical repetition may weaken students' initiative and creativity. The experimental group's teaching model transforms technical movement learning into an interesting experience through the introduction of smart classroom tools and gamified design. For example, the real-time feedback on hoop-throwing height and trajectory not only helps students adjust their movements precisely but also stimulates a sense of competition through visualizable achievements. Meanwhile, 3D movement decomposition and slow-motion replay reduce the cognitive load of complex skills, enabling students to gain the pleasure of exploration while understanding the principles of movements. In addition, the smaller standard deviation of the experimental group indicates that the new teaching method, while improving the overall interest level, does not significantly widen individual differences, which shows that it has high universality and inclusiveness.

However, the application of technical tools must balance the aesthetic characteristics of rhythmic gymnastics with the needs for personalized expression. Over-reliance on quantitative indicators may cause students to excessively focus on data performance while neglecting the artistry of movements, leading to potential conflicts between the "perfection" of hoop-tossing trajectories and the fluidity of dance rhythms. Future research needs to further explore strategies for balancing technical tools with artistic expressiveness—for example, generating personalized movement suggestions through AI or incorporating aesthetic evaluation dimensions into data analysis. Additionally, teachers' technical literacy and teaching design capabilities are key variables in technology-empowered teaching, necessitating strengthened faculty training to enhance their ability to integrate traditional experience with technical tools. It is recommended that follow-up studies expand the sample size, extend the intervention period, and explore hybrid teaching models that retain teachers' personalized guidance on artistic expression while using technology to optimize training efficiency, thus constructing a new teaching system that balances scientific rigor and artistic appeal.

**Discussion on students' satisfaction.** In rhythmic gymnastics teaching, training satisfaction is not only an important indicator for measuring teaching effectiveness but also a key driver for students' sustained participation and skill internalization. This study reveals the deep-acting mechanism of technological intervention on learning experience by comparing the impacts of traditional teaching and digitally empowered teaching on students' training satisfaction. As shown in Table 4, the mean satisfaction score of students in the experimental group (32.410) is significantly higher than that in the control group (28.440), and the difference reaches a significant level (T = −2.669, P = 0.010).This result suggests that the introduction of simulation technology not only improves skill acquisition efficiency but also enhances students' overall satisfaction by optimizing cognitive pathways and emotional experiences.

In traditional teaching, students' understanding of complex movements such as the "upward-thrown vertical hoop" mostly relies on teachers' verbal descriptions and demonstrations with limited times, which easily leads to cognitive load overload due to incomplete information transmission. The 3D movement simulation and real-time feedback system adopted by the experimental group transforms abstract movements into concrete modular information. This process conforms to the mechanism of "schema construction" in cognitive load theory students gradually establish a clear cognitive framework through the visual input of decomposed movements, thereby reducing the burden on working memory [15]. For example, the real-time data feedback on the hoop-throwing angle enables students to quickly identify movement deviations and form a closed-loop thinking of "problem - correction", thus improving the pertinence and efficiency of practice.

The improvement in satisfaction among the experimental group also stems from the emotional incentive effect empowered by technology. On one hand, instant positive reinforcement activates students' dopamine reward circuits, enhancing their learning motivation; on the other hand, the teaching experiment transforms training into competitive activities, stimulating students' sense of autonomy and control. This design aligns with the needs for "competence" and "autonomy" in self-determination theory, prompting students to shift from"passive imitation" to "active exploration". In addition, the diversity of technical tools may alleviate the boredom caused by trad"itional repetitive training. For example, simulating competition scenarios through virtual reality can enhance the immersion of training.

Although the satisfaction of the experimental group has significantly improved, its slightly higher standard deviation indicates that individual differences have not been completely bridged. This may be related to the adaptability threshold of technical tools: some students, due to technical operation obstacles or cognitive style preferences (such as field-dependent learners who rely more on teachers' guidance), find it difficult to benefit fully. Moreover, over-reliance on quantitative indicators, such as hoop-throwing accuracy, may lead students to neglect the aesthetic expression of rhythmic gymnastics. For instance, they might sacrifice the fluency and emotional tension of movements in pursuit of perfect data. Future research needs to explore the balance point of the mixed teaching model. For example, integrating aesthetic evaluation dimensions into technical analysis, or designing hierarchical technical tools to adapt to students with different learning styles [16]. At the same time, training for teachers' technology integration capabilities should be strengthened,

enabling them to play a bridging role between traditional experience and technical intervention, so as to avoid the risk of "dehumanization" in the teaching process.

## Conclusions and recommendations

### Conclusion

(1)  Relationship between projection angle and movement efficiency a vertical circle trajectory with a 30° projection angle demonstrates higher stability. Its moderate displacement effectively reduces cognitive load for beginners, making it suitable for standardized training of basic movements and short-term coordination. In contrast, a 45° projection angle increases airtime and rotation cycles, providing greater space for complex choreography and significantly enhancing the artistic expressiveness of movements. However, targeted training must be combined with students' strength levels.

(2) Differential effects of rotation directions clockwise rotation offers advantages in spatio-temporal consistency during apparatus tossing and catching, while counterclockwise rotation excels in movement transition smoothness. Teachers should flexibly select angles based on movement objectives and pay attention to how rotation directions affect technical efficiency. Combined training of the two rotation directions can expand movement diversity to meet the "innovative apparatus use" requirements in the FIG scoring system.

(3) Teaching empowerment of ADAMS technology students in the experimental group significantly outperformed those in the traditional teaching group in skill evaluation, interest in physical education contexts, and satisfaction. The teaching experiment results validate the reliability of the ADAMS model in predicting projection trajectories and joint torque distribution, providing a quantitative decision-making basis for teaching.

### Suggestions

As a benchmark tool in the field of multibody dynamics simulation, ADAMS demonstrates three core advantages in the teaching of rhythmic gymnastics: high-precision kinematic chain modeling, 3D trajectory visualization and reconstruction, and intelligent optimization of parameter sensitivity. This technology can shift rhythmic gymnastics teaching from experience-driven to data-driven, showing irreplaceable value in movement innovation, performance optimization, and other aspects. It is recommended that institutions gradually build an intelligent teaching closed loop of "simulation rehearsal-real-time correction-effect evaluation" and promote a dual-track model of "scientific quantitative guidance-artistic perceptual expression" in sports and art courses, ultimately achieving the collaborative improvement of technical precision and aesthetic value, the deep integration of sports science and difficulty-beauty sports projects, and the grand goal of innovative integration of artificial intelligence technology and education. In the future, further exploration can be conducted on the deep integration of ADAMS technology with VR immersive training to enhance students' spatial perception through three-dimensional virtual scenarios; meanwhile, a multimodal feedback system can be constructed by combining electromyographic signals and simulation data to provide more comprehensive technical support for the scientific and intelligent teaching of sports and art courses.

## Author contributions

**Conceptualization:** Zongjue Ma.

**Data curation:** Zongjue Ma.

**Writing – original draft:** Qihong Ren.

**Writing – review & editing:** Qihong Ren.

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
