## [Decision Letter · Decision Letter 0]

26 Mar 2025

Dear Dr. Ren,

Thank you for submitting your manuscript to PLOS ONE. After careful consideration, we feel that it has merit but does not fully meet PLOS ONE’s publication criteria as it currently stands. Therefore, we invite you to submit a revised version of the manuscript that addresses the points raised during the review process.

We look forward to receiving your revised manuscript.

Kind regards,

Dr. Guojin Qin

Academic Editor

PLOS ONE

**Journal Requirements:**

1. When submitting your revision, we need you to address these additional requirements. Please ensure that your manuscript meets PLOS ONE's style requirements, including those for file naming. The PLOS ONE style templates can be found at https://journals.plos.org/plosone/s/file?id=wjVg/PLOSOne_formatting_sample_main_body.pdf and https://journals.plos.org/plosone/s/file?id=ba62/PLOSOne_formatting_sample_title_authors_affiliations.pdf 2. Please note that PLOS ONE has specific guidelines on code sharing for submissions in which author-generated code underpins the findings in the manuscript. In these cases, we expect all author-generated code to be made available without restrictions upon publication of the work. Please review our guidelines at https://journals.plos.org/plosone/s/materials-and-software-sharing#loc-sharing-code and ensure that your code is shared in a way that follows best practice and facilitates reproducibility and reuse. 3. We note that the grant information you provided in the ‘Funding Information’ and ‘Financial Disclosure’ sections do not match.  When you resubmit, please ensure that you provide the correct grant numbers for the awards you received for your study in the ‘Funding Information’ section. 4. We note that your Data Availability Statement is currently as follows: All relevant data are within the manuscript and its Supporting Information files. Please confirm at this time whether or not your submission contains all raw data required to replicate the results of your study. Authors must share the “minimal data set” for their submission. PLOS defines the minimal data set to consist of the data required to replicate all study findings reported in the article, as well as related metadata and methods (https://journals.plos.org/plosone/s/data-availability#loc-minimal-data-set-definition). For example, authors should submit the following data: - The values behind the means, standard deviations and other measures reported;- The values used to build graphs;- The points extracted from images for analysis. Authors do not need to submit their entire data set if only a portion of the data was used in the reported study. If your submission does not contain these data, please either upload them as Supporting Information files or deposit them to a stable, public repository and provide us with the relevant URLs, DOIs, or accession numbers. For a list of recommended repositories, please see https://journals.plos.org/plosone/s/recommended-repositories. If there are ethical or legal restrictions on sharing a de-identified data set, please explain them in detail (e.g., data contain potentially sensitive information, data are owned by a third-party organization, etc.) and who has imposed them (e.g., an ethics committee). Please also provide contact information for a data access committee, ethics committee, or other institutional body to which data requests may be sent. If data are owned by a third party, please indicate how others may request data access. 5. Please amend either the title on the online submission form (via Edit Submission) or the title in the manuscript so that they are identical. 6. Please remove your figures from within your manuscript file, leaving only the individual TIFF/EPS image files, uploaded separately. These will be automatically included in the reviewers’ PDF. 7. Please upload a new copy of Figure 1, as the detail is not clear. Please follow the link for more information: "https://blogs.plos.org/plos/2019/06/looking-good-tips-for-creating-your-plos-figures-graphics/" https://blogs.plos.org/plos/2019/06/looking-good-tips-for-creating-your-plos-figures-graphics/

Reviewers' comments:

Reviewer's Responses to Questions

**Comments to the Author**

1. Is the manuscript technically sound, and do the data support the conclusions?

Reviewer #1: Partly

Reviewer #2: Yes

2. Has the statistical analysis been performed appropriately and rigorously?

Reviewer #1: Yes

Reviewer #2: Yes

3. Have the authors made all data underlying the findings in their manuscript fully available?

Reviewer #1: No

Reviewer #2: Yes

4. Is the manuscript presented in an intelligible fashion and written in standard English?

Reviewer #1: Yes

Reviewer #2: Yes

**Reviewer #1:**  1. Why only study 30 degrees and 45 degrees?

2.What is the reason for setting up these simulations?

3.Although there are a lot of simulation results, what is the difference with the actual performance?

4.The practical guiding significance of the article is not obvious and needs to be supplemented. At the same time, the difference between the simulation results and the experimental results also needs further analysis.

5.Please add a comparative analysis of the method in this paper and other related technologies in the discussion section.

**Reviewer #2:**  Review Comments on "Experimental Exploration of Artificial Intelligence ADAMS Simulation Technology in the Teaching of Vertical Hoop Thrown in Rhythmic Gymnastics"

Research Background: While the paper highlights the potential of artificial intelligence in sports education, it lacks a detailed analysis of the current state of rhythmic gymnastics teaching. It is recommended to supplement the discussion with a more thorough examination of existing challenges in rhythmic gymnastics education and how ADAMS technology can specifically address these issues. This will enhance the necessity and urgency of the research.

Literature Review: The literature review is relatively brief. It is suggested to expand this section by providing a more comprehensive synthesis of relevant studies, both domestic and international, particularly regarding the application of ADAMS technology in similar sports education contexts. This will help better position the study’s contribution to the field.

Research Methodology Details: In the process of constructing the ADAMS model, details regarding parameter settings and constraints of the rhythmic gymnastics hoop are insufficiently described. This lack of detail may hinder replication by other researchers. It is recommended to provide a step-by-step description of model construction along with key parameters to improve the reproducibility of the study.

Technical Details: The simulation process in ADAMS does not comprehensively account for certain physical factors such as air resistance and the elastic deformation of the hoop. It is suggested to elaborate on how these factors were considered in the simulation and their impact on the accuracy of the results.

Data Presentation and Analysis: Some figures are not clearly labeled, and the presentation of data could be further optimized. It is advised to refine the speed and displacement graphs to improve clarity and facilitate readers’ understanding of the results.

Depth of Discussion: While the overall structure of the paper is clear, the discussion on the impact of ADAMS technology on teaching effectiveness remains at a descriptive level. It is recommended to explore the underlying mechanisms in greater depth, such as students' cognitive processing and emotional engagement when using simulation technology, and how these factors collectively influence learning outcomes.

Implementation of Teaching Interventions: For the teaching interventions in both the experimental and control groups, it is advisable to provide a more detailed description of the specific activities, the use of teaching resources, and the role and actions of instructors. This will help readers better understand the differences in teaching approaches.

Effect Size Calculation and Interpretation: In addition to reporting significance test results, it is recommended to calculate effect sizes to provide a more comprehensive assessment of the impact of ADAMS simulation technology on teaching outcomes. Furthermore, the effect sizes should be interpreted in relation to real-world teaching contexts.

Conclusion Summary: The conclusion section provides a general summary of the findings but lacks specificity. It is advised to explicitly highlight the unique advantages of ADAMS technology in rhythmic gymnastics teaching and its practical application scenarios to offer more actionable insights for educators.

References: The references section could be diversified in terms of both quantity and variety. Increasing the number of cited studies and including a broader range of literature types would strengthen the academic foundation of the paper.

**Do you want your identity to be public for this peer review?** For information about this choice, including consent withdrawal, please see our Privacy Policy

Reviewer #1: No

Reviewer #2: No

---

## [Author Response · Author response to Decision Letter 1]

24 Jul 2025

Reply to the Editorial Board

1. The style of the article has been revised according to the specified format.

2. As this paper uses Adams Software to conduct virtual simulation research on the rhythmic gymnastics hoop, it should be noted that this software is a multi-body dynamics simulation software developed by the American company Mechanical Dynamics Inc. (later merged into MSC Software Corporation). As a commercial software with certain business confidentiality, it cannot be shared.

3. The affiliated project has been explained in the research topic, clarifying that this paper has not received financial support from the teaching reform project but is one of the achievement for the project's conclusion.

4.Data Availability Statement: All data generated in this study are included in the main text and supplementary materials, and no additional access is required. I confirm that my submission contains all the raw data necessary for the research results.

5. The title on the online submission form and the title in the manuscript have been revised to ensure they are identical.

6. Images have been deleted from the manuscript file, leaving only separate TIFF/EPS image files to be uploaded. These will be automatically included in the reviewers' PDF.

7. A new copy of Figure 1 has been uploaded to the article.

Responses to Reviewers' Questions:

1. The manuscript is technically reliable. The data generated by Adams Software provide scientific support for the paper's arguments and lead to authentic conclusions.

2. The authors have provided all data underlying the findings described in their manuscript, which are deposited in a public repository as supporting materials for the manuscript.

3. The answers to the above questions have been fully explained.

Reviewer #1:

Question 1:

First, in college rhythmic gymnastics teaching, 30-degree and 45-degree toss-and-catch are explicitly specified as primary and intermediate technical movements in the college rhythmic gymnastics teaching syllabus. These two angles cover the typical motion trajectory range of vertical toss-and-catch. The 30-degree angle leans more toward low toss with high control, suitable for stability training in the beginner stage; the 45-degree angle is close to the maximum parabolic distance, which can train students' strength and coordination. Focusing on this, the study aims to provide quantitative data directly aligned with the teaching syllabus for curriculum reform.

Second, according to projectile kinematics formulas, the differences in toss height (H) and horizontal displacement between 30 degrees and 45 degrees are significant. At 30 degrees, the vertical velocity component is smaller, and the rotational stability of the ring is higher; at 45 degrees, due to the close horizontal and vertical components, rotational deviation is prone to occur. In teaching, students often experience movement deformation (such as uneven wrist force). Through simulation comparison, targeted optimization of movement decomposition teaching can be achieved.

Finally, in actual teaching, students in the beginner stage struggle to precisely control toss-and-catch angles exceeding 45 degrees, easily leading to ring loss of control or sports injuries. The selection of 30 degrees and 45 degrees adheres to the teaching principle of "step-by-step progression" and can predict common errors through simulation, such as confusion in rotational direction.

Question 2:

First, rhythmic gymnastics toss-and-catch techniques rely on teacher demonstrations and student imitation, but the instantaneity of human movements makes it difficult for students to observe details, such as wrist angles and ring rotational angular velocity at the moment of projection. ADAMS simulation, through parametric modeling, can decompose movements into mechanical variables such as initial velocity and torque, providing visual slow-motion playback to compensate for the lack of dynamic perception in traditional teaching.

Second, the comparison between 30 degrees and 45 degrees reveals the influence of mechanical parameters on ring movement through differentiated projection angles. For example, 45-degree toss-and-catch requires higher horizontal initial velocity. Simulation can quantitatively show the premature fall of the ring caused by insufficient velocity, helping students understand the "angle-force" coordination mechanism.

Third, the innovation of introducing ADAMS multibody dynamics simulation, which is mature in industrial fields, into physical education lies in: (1) generating real-time feedback through parameter deviations between simulation results and standard movements to assist teachers in accurately locating student errors. (2) Promoting interdisciplinary curriculum design of "sports + engineering" to enhance students' scientific training literacy.

Question 3:

First, ADAMS simulation assumes constant air resistance, while in actual training, air turbulence, venue humidity, etc., may cause fine adjustments to the ring's motion trajectory. Simplifying environmental factors aids theoretical research.

Second, the simulation model is based on rigid-body dynamics, ignoring the flexible characteristics of the human musculoskeletal system. For example, the damping characteristics of wrist joints during students' toss-and-catch will affect the rotational stability of the ring, but the simulation assumes ideal rigid connections. In the future, flexible-body simulation can be introduced to calibrate the model.

Question 4: The practical guiding significance has been supplemented in the article, and the differences between simulation results and experimental results have been deeply analyzed.

Question 5: A comparative analysis of this method with other related technologies has been added to the discussion section.

Reviewer #2: Comments on "Experimental Exploration of AI ADAMS Simulation Technology in Vertical Ball Toss Teaching of Rhythmic Gymnastics"

Research Background: Detailed analyses of the current status, challenges, and specific problem-solving capabilities of ADAMS technology in rhythmic gymnastics teaching have been added.

Literature Review: The literature review has been expanded.

Methodological Details: The construction process of the ADAMS model in the research methods has been described in detail, with additional parameter settings and constraint descriptions.

Technical Details: The impact of ignoring air resistance and the elastic deformation of the rhythmic gymnastics ring on result accuracy in simulations has been clarified in the text of "3.1 Advantages of AI ADAMS Simulation Technology".

Data Presentation and Analysis: The line width in all velocity and displacement graphs has been adjusted from 1.5 points to 1 point to improve clarity.

Depth of Discussion: In response to this reviewer's comment, the following work has been done:

The discussion has been divided into two main parts: discussion of simulation results and discussion of experimental results, enhancing the paper's hierarchical structure and logical flow.

The depth and breadth of the discussion section have been expanded. In the discussion of simulation results, kinematic characteristics of different projection angles and teaching strategies, as well as the balance between projection angles and artistic expression, have been analyzed. In the discussion of experimental results, detailed discussions have been conducted on students' professional skills, interest in sports contexts, and satisfaction.

Implementation of Teaching Intervention: The implementation process of the teaching intervention has been described in detail in Section 4.4 (Discussion).

Calculation and Interpretation of Effect Size: The impact of ADAMS simulation technology on teaching outcomes has been analyzed by combining practical teaching contexts to evaluate effect size.

Conclusion and Summary: The conclusions and recommendations have been rewritten.

References: Due to the expansion of the literature review, a corresponding number of additional references of different types have been added.

---

## [Editor Report · Decision Letter 1]

7 Aug 2025

Experimental exploration of artificial intelligence and ADAMS simulation technology in the teaching of vertical hoop upward throw in rhythmic gymnastics

PONE-D-24-59239R1

Dear Dr. Ma,

We’re pleased to inform you that your manuscript has been judged scientifically suitable for publication and will be formally accepted for publication once it meets all outstanding technical requirements.

Kind regards,

Guojin Qin

Academic Editor

PLOS ONE
---

## [Editor Report · Acceptance letter]

PONE-D-24-59239R1

PLOS ONE

Dear Dr. Ma,

I'm pleased to inform you that your manuscript has been deemed suitable for publication in PLOS ONE. Congratulations! Your manuscript is now being handed over to our production team.

Kind regards,

on behalf of

Dr. Guojin Qin

Academic Editor

PLOS ONE